# Rainfall-Runoff Time Lags from Saltwater Interface Interactions in Atlantic Coastal Plain Basins

**Brady Evans** [1], **Harald Klammler** [1,2], **Michael D. Annable** [1,*] and **James W. Jawitz** [3]

1 Engineering School of Sustainable Infrastructure and Environment (ESSIE), University of Florida, Gainesville, FL 32611, USA
2 Department of Geosciences, Federal University of Bahia, Salvador 40170, Brazil
3 Soil and Water Sciences Department, University of Florida, Gainesville, FL 32611, USA
* Correspondence: annable@ufl.edu

**Abstract:** The dynamic behavior of the freshwater-saltwater interface (FSI) in coastal aquifers can introduce unexpected lags between recharge and stream discharge, especially when recharge is forced by long-term cyclical precipitation patterns. This work seeks to assess these FSI impacts at the watershed scale. Recharge-discharge time lags were evaluated in 68 watersheds overlying the Floridan Aquifer System in the coastal region of the southeastern United States (Florida, Georgia, and South Carolina). Utilizing the strength of the Atlantic multidecadal oscillation (AMO) signal in this region, 10–20 year averaged recharge and discharge time series were used for the selected watersheds. Lags of 10–25 years between recharge and discharge were found in 16% of the basins considered, possibly induced by a dynamic FSI which responded slowly to the AMO-scale recharge signal. Freshwater storage coefficients ($S$) were estimated from time series of change-in-storage and groundwater level, with 11 basins showing $S > 1.5$ indicating water storage well above that expected for unconfined aquifers. These 11 basins with both multidecadal recharge-discharge time lags and high $S$ values showed a positive linear relationship between time lag and FSI depth with slope 0.016 yr/m (R-squared = 0.30). These large time lags may be directly impacting the management of these basins as they obscure water and solute mass balances in the southeastern US.

**Keywords:** saltwater intrusion; water budget; watershed assessment

## 1. Introduction

Fresh groundwater is a vital resource in coastal regions [1,2]. Aquifers in these coastal regions face increasing vulnerability from environmental stressors such as population growth and migration, freshwater loss due to sea level rise, long-term changes in recharge, and saltwater surges and inundation [2–5]. The management of fresh groundwater resources in these regions depends on robust predictions of these transient phenomena and their impacts on the water balance and fresh groundwater storage.

The presence of a freshwater-saltwater interface (FSI) in coastal aquifers affects the dynamic response of the freshwater balance to cyclical climate variability at timescales corresponding to daily tidal cycles [6–8], seasonal recharge oscillations [9–12], and Atlantic multidecadal oscillation (AMO) [13]. Seasonal oscillations in recharge ($R$) are reflected in measured or modeled submarine groundwater discharge at the coastline, but with time lags ($\Delta t$) of up to five months attributed to the delayed response of the FSI to the $R$ signal [10,14]. Likewise, AMO scale $R$ leads to multidecadal oscillations in stream discharge [15] which may lag $R$ on the order of 15 years [13]. The time required to move large volumes of freshwater and saltwater depends on both the periodic $R$ signal wavelength [13] and the aquifer hydraulic response time $T_r$ [16]

$$T_r = \frac{L^2 S}{T} \tag{1}$$

where $L$ is a characteristic aquifer length scale [L], $S$ is the storage coefficient [-], and $T$ is transmissivity [$L^2$/T]. The response time represents the lag in the baseflow ($Q_b$) response to the oscillating $R$ signal.

During the increasing phase of an oscillating $R$ signal, the difference in freshwater and saltwater densities causes the FSI to migrate downward approximately 40 times the increase in water table elevation $h$ (Ghyben-Herzberg relation, Figure 1A,B). A consequence of this significant FSI movement is that increases in $R$ translate more to the displacement of saltwater in the aquifer than to increasing $Q_b$, thus leading to a delayed $Q_b$ signal. The second consequence of this FSI response is that changes in freshwater storage in coastal aquifers primarily occur below sea level, with changes in $h$ reflecting only ~2.5% (1/41) of the total change in freshwater storage, $\Delta V$ (illustrated in Figure 1C based on the integration of the difference between recharge $R$ and baseflow $Q_b$). Accurate estimation of freshwater storage in coastal aquifers thus requires consideration of FSI movement. Neglecting this process may result in an underestimation of the total change in freshwater storage by up to a factor of 41 following a change in $R$ [17].

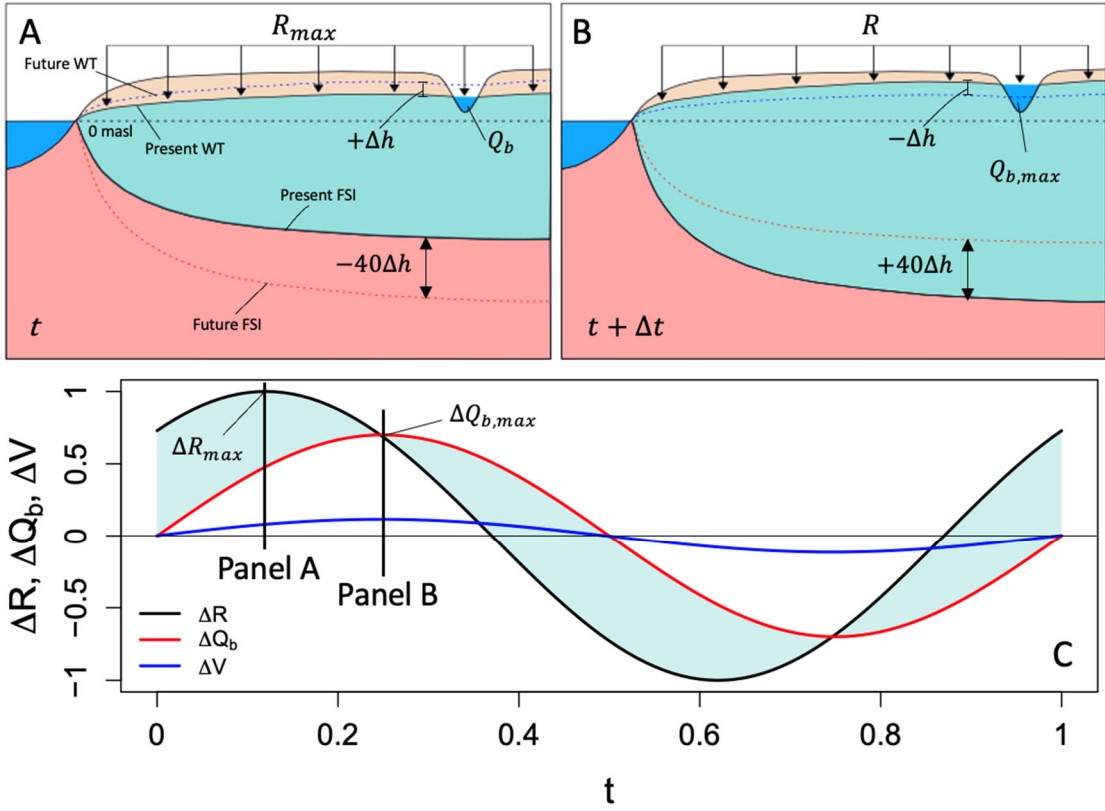

**Figure 1.** An example of the lagged response of freshwater (light blue), saltwater (light red), and inland baseflow discharge ($Q_b$) to an oscillating recharge ($R$) signal in regions with the good hydraulic connection between FSI and the water table (WT). (**A**) During maximum $R$, the FSI must move down by $40\Delta h$ until (**B**) maximum $Q_b$ occurs at $t = t_0 + \Delta t$ where the FSI must move up by $40\Delta h$ after the decrease in $R$. (**C**) Integration of $R - Q_b$ (black line minus red line) with respect to time shows the periodic change in freshwater storage ($\Delta V$) in blue. Changes in both FSI and WT lag the $R$ signal, and the corresponding current vs. future FSI and WT are shown in panels A and B as dotted lines (red and blue, respectively) compared to the current levels shown as solid black lines.

The long-term storage dynamics shown in Figure 1C can provide information related to the hydrogeologic connection between surficial aquifers and aquifers containing the FSI. These storage dynamics can be quantified using $S$ as the reconciliation factor for the observed changes in groundwater levels ($\Delta h$) with the calculated changes in $\Delta V$. This estimation is consistent with the definition of specific yield. Due to the fact that freshwater

storage near the FSI is impacted by variable density effects, the relation $S = 40\eta$ (where $\eta$ is porosity near the FSI) is expected in basins impacted by FSI dynamics [13].

The magnitude of the delayed response of the FSI and subsequent aquifer discharge can be expected to increase with the distance of the discharge point from the coast, as reflected by $L^2$ in Equation (1). However, the specific hydrogeological conditions in which the FSI can be responsive enough to produce a lagged $Q_b$ response to a changing $R$ condition are unknown. In the major karst springshed of Silver Springs Florida, spring discharge was observed to lag recharge by approximately 15 years over a multidecadal timescale, with FSI interactions as a hypothesized driver [13]. Those authors found that $\Delta t$ occurred in model scenarios with and without a semi-confining unit (modeled using a hydraulic resistance parameter) between the recharge boundary and the FSI, suggesting that the FSI can influence the dynamics of the upper aquifer's freshwater balance even when there is some degree of hydraulic separation between the two. However, the spatial extent of this phenomenon across different catchments on the multidecadal scale remains unknown. This work seeks to assess the widespread impact of the FSI on multidecadal freshwater balances at the watershed scale and to determine potential hydrogeologic conditions in which the FSI can lead to lagged $Q_b$.

The southeastern United States (SEUS) was selected as a study region because the climate and hydrogeology are hypothesized to be conducive to recharge-discharge time lags. The deeper formations of the carbonate-rock Floridan Aquifer System (FAS) in the SEUS have the potential to contain saltwater, especially in Georgia and Florida [18–21]. Georgia has a deep, extensive, highly permeable zone that is mostly saline, with a fractured overlying confining layer that provides a hydraulic connection to the FAS [22]. Likewise in Florida, the well-established hydraulic connection between the mostly saline Lower Floridan Aquifer (LFA) and the Upper Floridan Aquifer (UFA) [23–25] suggests that the FSI could migrate in response to head changes experienced in the UFA as the pressure propagates through the LFA. Additionally, the AMO index has been described as the strongest climate index in this region [26], and the SEUS also shows the strongest connection between streamflow ($Q_s$) and AMO when compared with other US regions [27]. Given the hydrogeologic information linked to SEUS and the strength of multidecadal signals in rainfall and $Q_s$, this study focused on streams and springs with the potential for $Q_b$ lags in both Florida and coastal Georgia.

Areas with potential for a lagged $Q_b$ response relative to $R$ were hypothesized to have a good hydraulic connection (i.e., without low permeability units) between the FSI and overlying portions of the aquifer reacting to a multidecadal $R$ signal. This hypothesis was tested using recharge and runoff data from basins that clearly fit these criteria and also basins that do not have an FSI. Also, considered are basins in which FSI connectivity may be heterogenous. The testable predictions from this hypothesis are that basins without hydraulic connection to FSI were expected to have no observable lag and low $S$ values, while hydraulic connection to FSI should show lagged $Q_b$ and much larger $S$ values. The threshold value of $S > 1.5$ was used in this study given that $\eta$ in carbonate rocks is most often greater than 4% at depths shallower than 2 km [28]. Here both streamflow ($Q_s$) and baseflow ($Q_b$) were evaluated, with lagged $Q_s$ due to FSI storage expected in groundwater-dominated basins whereas $\Delta t$ uniquely in $Q_b$ and not in the total $Q_s$ was expected in basins with a reduced groundwater contribution. In either case, the discovery of $\Delta t$ occurring in multidecadal groundwater discharge signals in the SEUS as a component of $Q_s$ has long-term implications for the management of water and solute balances of this region.

## 2. Methods

### 2.1. Hydrologic Data

Water budget components needed for estimating long-term $\Delta t$ between long-term $R$ and $Q_b$ include precipitation, evapotranspiration (ET), surface runoff ($Q_{ro}$), and ground-

water pumping ($Q_p$). Long-term average $R$ and $Q_b$ were assumed to be equivalent to the water budget

$$R = P - ET - Q_p - \varepsilon = Q_b \tag{2}$$

where $\varepsilon$ is the mass balance error in each watershed, and with $Q_b$ derived from $Q_s$ data

$$Q_s - Q_{ro} = Q_b \tag{3}$$

Data were compiled for 68 perennial streamflow stations that met the following criteria: more than fifty years of daily discharge data, less than three years of data gaps, and in areas with evidence of a saltwater presence in the aquifer. Based on data available from USGS [29] (number of basins (n) = 52), SJRWMD [30] (n = 2), and SRWMD [31] (n = 1), fifty stations in Florida and five in Georgia fit these criteria. Thirteen other stations in Georgia, North Carolina, and South Carolina that lacked evidence of saltwater presence were selected under these criteria to compare with the 55 basins with evidence of a saltwater presence. Daily stream discharge data were resampled for these 68 stations to monthly resolutions because this study is focused on long-term sinusoidal signals in the data. The influence of surface runoff on time lag estimation was removed with baseflow separation of the daily stream discharge data (Equation (3)) using a recursive digital filter using the forward-backward-forward pass system with filter parameter 0.925 recommended by Nathan & McMahon [32].

Monthly rainfall data from 1895 to the present were obtained from the Parameter-elevation Relationships on Independent Slopes Model [33] at the centroids of the 68 basins considered in this study. Potential evapotranspiration (PET) in the humid subtropical climate of the study area was estimated using the Turc equation assuming relative humidity > 50% [34] with daily solar radiation estimated from Hargreaves & Samani [35], empirical radiation coefficient estimated from Samani [36], extraterrestrial radiation via Allen et al. [37], and daily minimum and maximum temperature data. ET was then estimated from the Choudhury [38] equation using the default value of 1.8 for the landscape parameter. Daily temperature data from 69 land-based stations (https://www.ncdc.noaa.gov/, accessed on 12 January 2021) were selected based on proximity to the drainage basins, data completeness, and length of the record. Data completeness in 53 of the 69 stations was greater than 90%. There were 50 stations that had more than 80 years of data and the other 19 stations were primarily used to interpolate missing values. Prior to interpolation, the daily temperature was converted to the monthly average temperature. The interpolation method in this study used an algorithm to find both the nearest and most highly correlated temperature data stations when filling in time series gaps (Supplementary Materials). Groundwater pumping rates were estimated every five years in the period 1950–2015 using state-level groundwater usage data from USGS [39], which were also linearly extrapolated to 2020. The $Q_p$ time series for each basin was determined using the proportion of each basin's drainage area to the approximate area of each state to convert the state-level data to the watershed scale.

Aquifer head data was obtained from USGS [39] (n = 66 basins), SJRWMD [40] (n = 1), and SWFWMD [41] (n = 1) based on the length of record and locations relative to the basin area. All wells with greater than 30 years of data inside each basin's boundary were considered for analysis (ranging from zero to four wells per basin). In cases with no wells (n = 17), up to two wells outside of but nearby a basin's boundary was selected. Three basins considered in this study did not have any wells within or nearby the basin's boundary with long records and were thus not considered for this analysis.

### 2.2. Time Lag Estimation

The presence or absence of a $\Delta t$ between long-term $R$ and $Q_b$ oscillations were tested using crosscorrelation analysis. Moving average windows were used to remove higher frequencies present in the $R$ and $Q_b$ signals. Cross correlation was evaluated with moving windows of 10 to 20 years in increments of one year for each basin, consistent with studies

linking long-term $Q_s$ trends to multidecadal climate indices which use both 10 year [15,42] and 20 year [43–45] moving windows. After applying a moving window to both $R$ and $Q_b$ time series, the averaged $Q_b$ time series was translated back in time in increments of one month until the beginning of the typically longer $R$ time series was reached. The interval that corresponded to the maximum correlation between $R$ and the lagged $Q_b$ values was recorded as $\Delta t$ (e.g., Figure S1) for all 11 moving average windows. The first condition for evidence of FSI impacts to the water balance in each basin was evaluated as mean $\Delta t > 1$ yr with the coefficient of variation ($CV$) of the 11 $\Delta t$ estimates < 0.3 (e.g., Table S2). Basins with mean $\Delta t < 1$ yr or $CV > 0.3$ were considered unlikely to be impacted by the FSI.

*2.3. Aquifer Storage Estimation*

The $\Delta V$ time series was determined as the cumulative difference between $R$ and $Q_b$ divided by drainage area. To eliminate water balance errors ($\varepsilon$ in Equation (2)) not caused by FSI dynamics, the long-term mean of $R$ was matched with that of $Q_b$ prior to creating the $\Delta V$ time series. This methodology resulted in a cumulative change in storage between $-0.008$ and $0.02$ m/yr for all basins. Both $\Delta h$ and $\Delta V$ time series were averaged with a 10-yr moving window, truncated to include coinciding dates, and shifted to long-term mean zero. Lastly, $S$ was estimated by maximizing the Nash-Sutcliffe efficiency (NSE) between $\Delta V$ and $\Delta h$ using the *optimize* function in RStudio (v. 1.0.153) with limits of $S$ set to (0.50) (e.g., Figure S2). Implicit in this approach is the assumption that there is a good hydraulic connection between the WT and the FSI. Low values of either NSE or $S$ may indicate a lack of this hydraulic connection.

Similar to $\Delta t$, estimates of $S$ were distributed into three categories: no saltwater interaction (NSI), evidence of some saltwater interaction (SSI), and indicative of saltwater interaction (SI). This categorization ultimately depends on the aquifer's specific yield and its porosity near the FSI for a given basin [13]. In this study, $S > 1.5$ indicated SI (implying a lower aquifer porosity of 4% or larger based on $S = 40\eta$), $S < 0.4$ were interpreted as NSI because upper limits of specific yield in various geologic materials are typically between 35–40% [46,47], and values between these limits ($0.4 < S < 1.5$) were considered SSI (e.g., Table S3). Thus, the second condition for evidence of FSI impacts on the water balance in each basin was $S > 0.4$. Only basins that satisfy both conditions for FSI impacts ($\Delta t > 1$ yr with $CV < 0.3$ and $S > 0.4$) were impacted by the dynamic behavior of the FSI.

*2.4. Basin Delineation and Hydrogeology*

Most basins (46) in the study were delineated by USGS [48] when determining the upstream drainage areas of streamflow gaging stations. The USGS Watershed Boundary Dataset (WBD) was used to delineate 11 other basins. In these cases, each delineated drainage area upstream from a gaging station was merged into one larger basin. Details on the delineations of eight other basins can be found in the supplementary information.

The 13 basins without saltwater presence cover regions that overlie the FAS, the semi-consolidated sand aquifers of the Atlantic coastal plains, and the igneous and metamorphic rock aquifers of the Piedmont and Blue Ridge crystalline-rock aquifer system (Figure 2). Three of the 55 basins with saltwater presence are first magnitude springs, two basins are second-magnitude springs, and the remaining 50 basins are streams and rivers residing over the carbonate-rock aquifers of the FAS. These aquifers exhibit a wide range of transmissivity ($T$) from 10 m$^2$/d to over 10$^6$ m$^2$/d in highly karstified areas [21]. Most of the FAS is overlain by an upper confining unit (UCU) but is unconfined in north-central Florida ("Carbonate-rock aquifers" area in Figure 2). Typical hydrogeological cross-sections in the study area contain a thin clay UCU over the freshwater UFA with the mostly saline LFA lying 400–600 m below the ground surface and leaky or non-leaky middle confining units between the LFA and the UFA. For detailed cross-sections in specific regions of the study area, the reader is referred to Williams and Kuniansky [21] and Plate 30 in Miller [20].

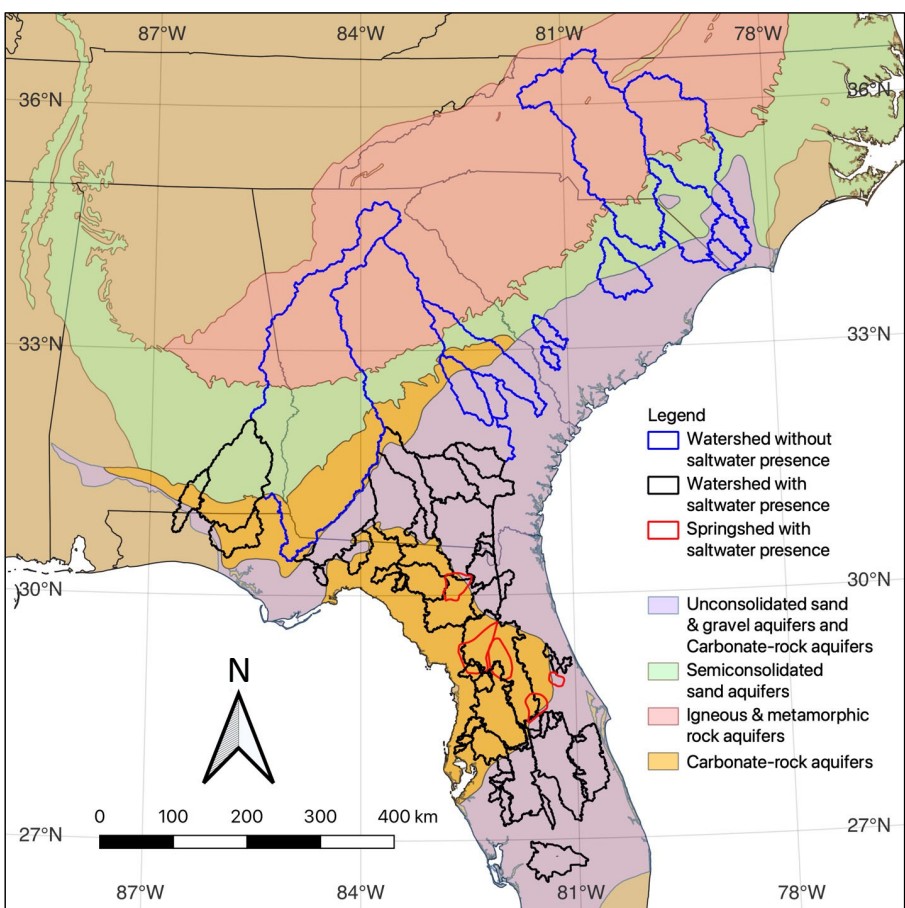

**Figure 2.** The 55 watersheds with saltwater presence (black and red outlines) are in areas with carbonate-rock aquifers and unconsolidated sand and gravel aquifers, and the 13 watersheds without saltwater presence (blue outlines) are in areas with similar aquifers in addition to igneous and metamorphic rock aquifers.

## 3. Results and Discussion

### 3.1. Methodology Validation

The 13 basins without evidence of saltwater presence were each concluded to not be impacted by the dynamic behavior of the FSI using the outlined methodology for $\Delta t$ estimation and $S$ estimation (Table 1). As explained in the introduction, the dynamic response of the FSI to changes in recharge leads to changes in storage that are approximately 40 times the observed change in head. This concept indicates $S = 40\eta$ in basins impacted by FSI dynamics. Thus, the 13 basins without evidence of saltwater presence were expected to have $S$ less than or equal to typical values of porosity. Six of these thirteen basins had $S < 10^{-4}$ with NSE < 0, which suggests a lack of hydraulic connection between the potentiometric surface and the lower aquifer because the $\Delta h$ time series did not correlate well with the $\Delta V$ time series. Two of the thirteen basins did not have groundwater data, four basins had $S < 0.4$, and one basin had $S = 0.54$. The aquifer storage estimation methodology thus indicates one of these thirteen basins could be impacted by the FSI. However, the $\Delta t$ estimation methodology for this basin suggested there is likely no lag present in this basin (Basin 43 in Table 1). Due to the fact that none of the 13 basins without evidence of saltwater presence satisfied both the $\Delta t$ and $S$ conditions for evidence of FSI impacts, all 13 basins were considered as correctly analyzed by the methodology used in this study. Thus, the remainder of this analysis only considers the 55 basins with evidence of saltwater presence.

**Table 1.** Estimations of $\Delta t$ and $S$ produced from the methodology used in this work for the 13 basins without evidence of saltwater presence. The symbols $\Delta t_\mu$ and $\Delta t_\sigma$ are mean time lag and standard deviation of estimated time lags, respectively.

| | S Estimation | | $\Delta t$ Estimation | | |
|---|---|---|---|---|---|
| **Basin** | $S$ | **NSE** | $\Delta t_\mu$ **(yr)** | $\Delta t_\sigma$ **(yr)** | $CV=\frac{\Delta t_\sigma}{\Delta t_\mu}$ **(-)** |
| 2—Altamaha River, GA | $7.0 \times 10^{-5}$ | $-2.0 \times 10^{-4}$ | 5.87 | 8.03 | 1.4 |
| 4—Apalachicola River, FL | 0.085 | 0.67 | 15.5 | 1.32 | 0.085 |
| 5—Black River, SC | 0.10 | 0.56 | 1.20 | 0.496 | 0.41 |
| 9—Canoochee River, GA | $7.0 \times 10^{-5}$ | $-3.0 \times 10^{-4}$ | 0.0833 | 0.00 | 0.0 |
| 10—Cape Fear River, NC | $7.0 \times 10^{-5}$ | $-3.0 \times 10^{-5}$ | 9.61 | 9.48 | 0.99 |
| 13—Coosawhatchie River, SC | 0.39 | 0.81 | 0.485 | 0.517 | 1.1 |
| 27—Little Pee Dee River, SC | $7.0 \times 10^{-5}$ | $-7.0 \times 10^{-6}$ | 2.69 | 1.51 | 0.56 |
| 35—Ogeechee River, GA | 0.20 | 0.59 | 0.439 | 0.218 | 0.50 |
| 36—Ohoopee River, GA | $7.0 \times 10^{-5}$ | $-4.0 \times 10^{-4}$ | 0.0909 | 0.0251 | 0.28 |
| 38—Pee Dee River, SC | $7.0 \times 10^{-5}$ | $-6.0 \times 10^{-5}$ | 14.0 | 1.27 | 0.091 |
| 43—Salkehatchie River, SC | 0.54 | 0.69 | 6.00 | 5.11 | 0.85 |
| 61—Waccamaw River, NC | - | - | 15.3 | 12.0 | 0.78 |
| 62—Waccamaw River, SC | - | - | 12.0 | 12.0 | 1.0 |

In addition to the 13 basins without evidence of saltwater presence, the methodology was validated using results from Klammler et al. [13] who showed Silver Springs in north-central Florida had a $\Delta t = 14.9 \pm 0.5$ yrs and $S = 3$ using an analytical linear reservoir model allowing one-dimensional movement of the FSI in response to a multidecadal recharge signal. The authors also used an axisymmetric numerical model of the springshed which showed $\Delta t = 16$ yrs. The methodology used in the present study led to $\Delta t = 18.8 \pm 0.4$ yrs and $S = 2.5$, which suggests a potential overestimation of $\Delta t$. However, both estimations of $\Delta t$ and $S$ were of similar scale between this work and Klammler et al. [13], which suggests the methodology in this work can identify basins impacted by the FSI but may not produce accurate estimates of $\Delta t$.

### 3.2. Time Lag Estimation

Estimated $\Delta t$ values for the 55 basins were positively skewed with median $\Delta t = 5.65$ years and maximum $\Delta t = 30$ years (Figure 3). More than half of the basins (64%) showed $\Delta t > 1$ year, which suggests that most basins over aquifers with a saltwater presence may show lagged $Q_b$ in relation to $R$ on the multidecadal scale. It is important to note the values of $\Delta t$ estimated for the 55 basins are not constant in time. Rather, the rate of change in storage and therefore the response time is controlled by $R$ [5,49]. The multidecadal oscillations in $R$ examined in this study initiate FSI response and lead to lagged $Q_b$, but changes in long-term average $R$, for example due to anthropogenic climate change [50,51], would impact the value of $\Delta t$. Long-term water balance predictions for specific basins thus should account for this potential impact of $R$ on $\Delta t$. In this study, the values of $\Delta t$ estimated were used simply to observe the range of the delayed $Q_b$ responses.

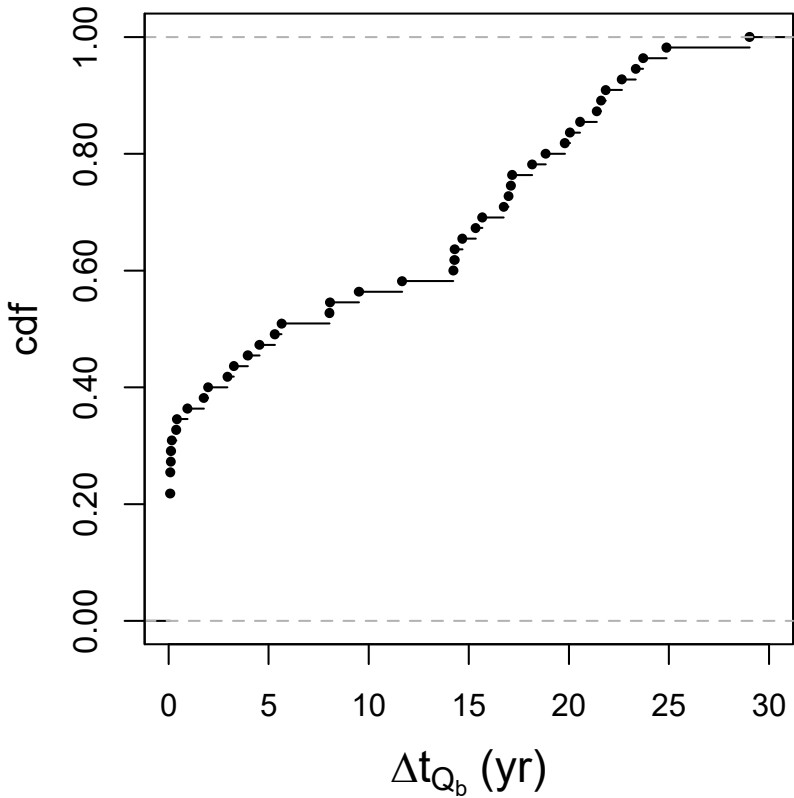

**Figure 3.** Cumulative distribution (CDF) of mean $\Delta t$ between $R$ and $Q_b$ for all 55 basins with saltwater presence, with 36% showing $\Delta t < 1$ year and the remaining 64% between 1–30 years.

Estimated values of $\Delta t$ differed between using $Q_b$ and $Q_s$. There were 14% fewer basins with $\Delta t > 1$ when $\Delta t$ was estimated using $Q_s$ compared to using $Q_b$. When comparing the mean $\Delta t$ for both $Q_s$ and $Q_b$ (Figure 4), 55% of basins showed $Q_b$ lag > $Q_s$ lag, 27% of basins showed $Q_b$ lag < $Q_s$ lag, and the $Q_b$ lag was equal to $Q_s$ lag in 18% of the 68 basins. The slope of a regression line was 1.2, supporting the observation of longer $\Delta t$ with $Q_b$. This observed increase in $\Delta t$ from $Q_s$ to $Q_b$ suggests that surface water runoff interfered with the $\Delta t$ estimation and that $Q_b$ is more likely to show a $\Delta t$ in a basin compared to $Q_s$. Due to the fact that there is a mix of high surface runoff watersheds and high $Q_b$ watersheds in the study area [52], some watersheds considered may contain a magnitude of surface runoff that could mask the effect of delayed $Q_b$ caused by FSI dynamics. Thus, the higher likelihood of measuring a long $\Delta t$ with $Q_b$ rather than $Q_s$ is consistent with expectations.

Lagged baseflow implies lagged solute delivery to these surface water systems, highlighting their importance for planning and management of water quality in these systems. Future studies interested in estimating a discharge $\Delta t$ induced by FSI dynamics in a surface watershed should use baseflow separation on the $Q_s$ time series to determine this $\Delta t$. The automated baseflow separation procedure used for all 68 basins in this study is common in the literature, but other methods may be preferred for individual basins based on local catchment characteristics [53] such as hydraulic conductivity ($K$) and the ratio of precipitation to potential evapotranspiration (P/PET) [54].

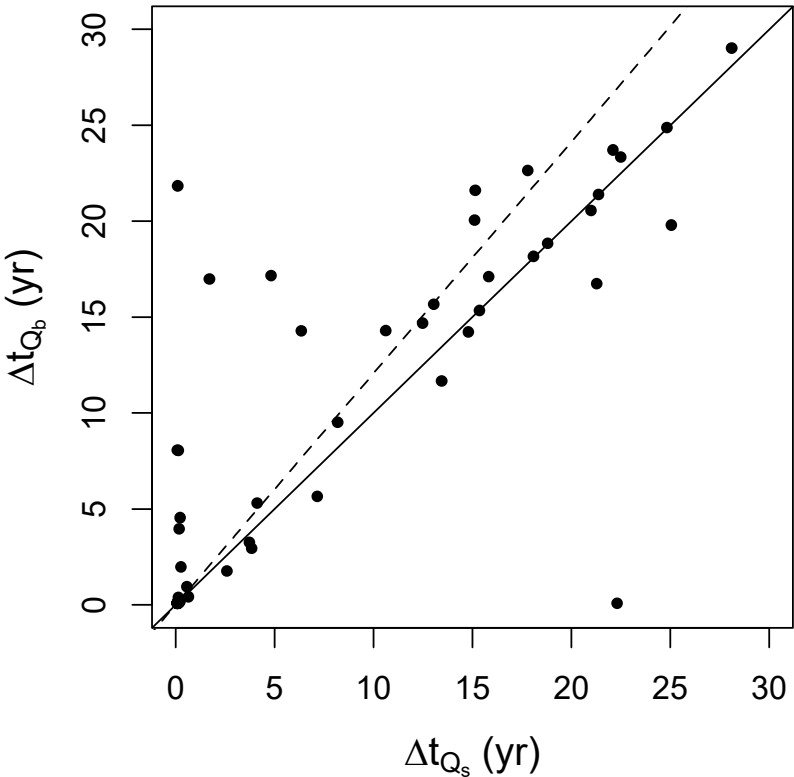

**Figure 4.** Mean $\Delta t$ between recharge ($R$) and baseflow ($Q_b$) plotted against mean $\Delta t$ between $R$ and streamflow ($Q_s$). Points above the solid line indicate $\Delta t_{Q_b} > \Delta t_{Q_s}$ in a basin. The dashed line is the best-fit linear regression with a slope of 1.2, $R^2 = 0.81$ (note that there are 19 points between (0, 0) and (1, 1)).

*3.3. Groundwater Storage Coefficient Estimation*

Groundwater storage coefficients were estimated successfully in 85% of the 55 basins considered, while in the remaining basins mass balance calculation led to low $S$ values approaching zero (particularly in basins with low correlation between $\Delta h$ and $\Delta V$). Low values of $S$ suggest that groundwater levels accounted for the observed changes in storage. For basins with evidence of a saltwater presence, low $S$ values suggest there is a poor hydraulic connection between the WT and FSI. Thus, eight basins with $S$ approaching zero were considered to not be impacted by the FSI.

Approximately half of the 55 basins showed at least some evidence of saltwater interaction, with 29% SSI (n = 16) and 24% SI (n = 13, Figure 5). Of the 18 NSI basins, 14 contain a confining unit between the UFA and the saltwater-bearing LFA, consistent with expectations that a hydraulic connection between freshwater and saltwater is required to generate $S > 1.5$. The six basins with the strongest evidence of FSI, with both $\Delta t > 10$ years and $S > 1.5$, are listed in Table 2. Basins 39, 40, 48, and 63 are in the unconfined, highly karstified portions of the FAS, while basin 49 is a short distance west of the Woodville Karst Plain in a confined area of the FAS and basin 59 overlies the thinly confined portion of the FAS (Figure 6). This mixture of unconfined and confined conditions in these six basins with both $\Delta t > 10$ years and $S > 1.5$ suggest that the FSI may impact $Q_b$ on the multidecadal scale even in cases with significant hydraulic resistance between the FSI and the water table.

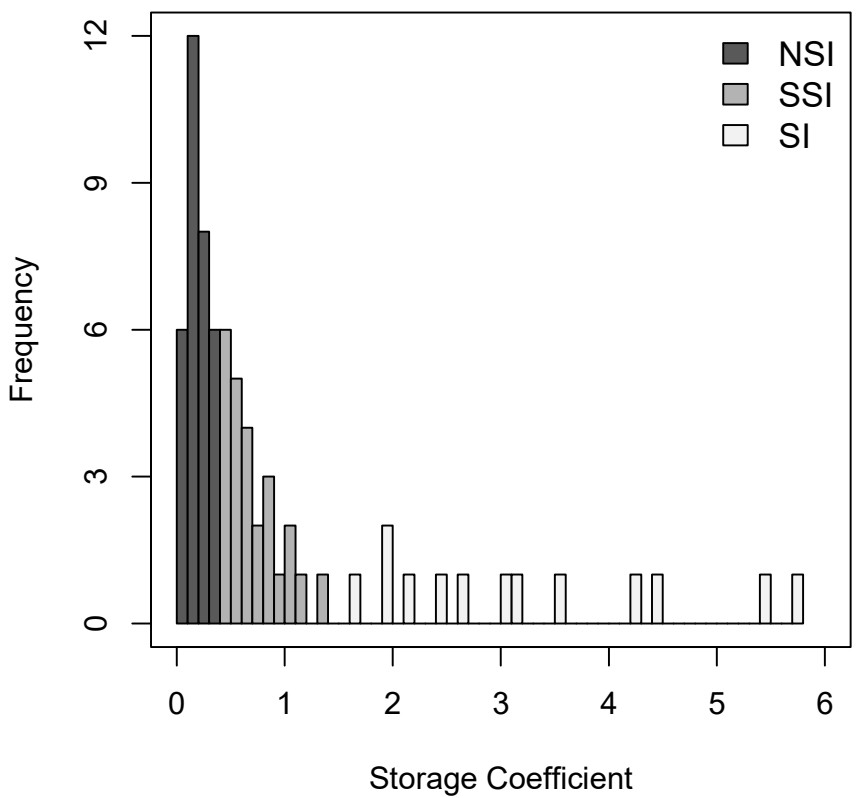

**Figure 5.** Distribution of estimated storage coefficients using the $\Delta h$ and $\Delta V$ time series. Note that NSI storage coefficients are 0–0.4, SSI storage coefficients are 0.4–1.5, and SI storage coefficients are >1.5.

**Table 2.** Eleven basins showing $\Delta t > 10$ yrs and either SI or SSI ($S > 0.4$), ordered from highest to lowest estimated $S$. FSI Depth is interpreted from Figure 54 in Williams and Kuniansky [21]. Confinement describes the percent of a basin's area overlying an unconfined (U), thinly confined (TC), or confined (C) aquifer. MCU Presence describes the basin area percent overlying a confining unit separating the UFA from the LFA; MCU I is leaky, MCU II is not leaky, and MCU III is somewhat leaky. Both confinement and MCU presence are determined from FAS maps in Miller [20].

| Basin No. | $\Delta t$ (yr) | $S$ | Area (km$^2$) | FSI Depth (m) | Confinement | MCU Presence |
|---|---|---|---|---|---|---|
| 49—Sopchoppy River, FL | $11.7 \pm 0.37$ | 5.74 | 275 | 150 | 100% C | None |
| 39—Pithlachascotee River, FL | $14.7 \pm 1.0$ | 5.44 | 410 | 335 | 43% U, 57% TC | 100% MCU II |
| 40—Rainbow River, FL | $21.4 \pm 1.0$ | 4.41 | 1920 | 520 | 61% U, 39% TC | 52% MCU II |
| 59—Tomoka River, FL | $21.8 \pm 1.1$ | 3.17 | 155 | 200 | 100% TC | 100% MCU I |
| 48—Silver River, FL | $18.8 \pm 0.42$ | 2.46 | 1280 | 500 | 86% U, 14% TC | 97% MCU I, 15% MCU II |
| 63—Withlacoochee River, FL | $24.9 \pm 1.4$ | 2.14 | 3980 | 600 | 68% U, 31% TC, 1% C | 57% MCU I, 97% MCU II |
| 30—North Fork Black Creek, FL | $23.7 \pm 1.1$ | 1.14 | 450 | 450 | 100% C | 100% MCU I |
| 67—Wolf Creek, FL | $17.1 \pm 1.4$ | 0.802 | 85 | 500 | 100% C | 100% MCU I |
| 16—Fenholloway River, FL | $21.6 \pm 1.3$ | 0.508 | 265 | 450 | 100% U | 49% MCU III |
| 53—St Marys River, FL | $20.1 \pm 1.0$ | 0.466 | 1850 | 550 | 100% C | 9% MCU I, 2% MCU III |
| 21—Ichetucknee Springs, FL | $15.3 \pm 2.0$ | 0.432 | 990 | 425 | 15% U, 60% TC, 25% C | 88% MCU III |

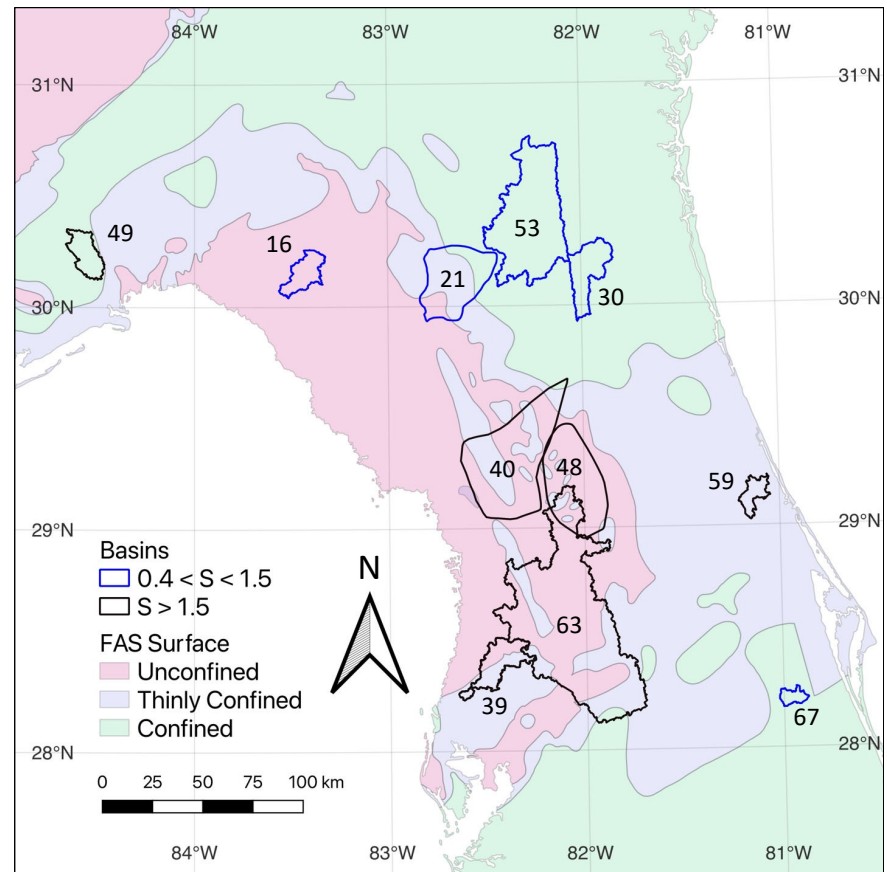

**Figure 6.** Map of SI basins (black outline) and SSI basins (blue outline) with significant Δ*t*. The map's base layer is shaded according to the confinement of the FAS—pink is unconfined, blue/purple is thinly confined, and green is confined. Note that SI basins tend to overlie unconfined or semi-confined conditions whereas SSI basins tend to overlie confined conditions.

For the 13 SSI basins, it may be appropriate to estimate an average *S* using specific yield for NSI areas and FSI storage (40*η*) for SI areas. Area-weighting may be sufficient in springsheds in which there is a single discharge point, but a more complex methodology may be necessary for a surface watershed with groundwater discharge throughout the stream network. Development of these methodologies was beyond the scope of this work but may be important for understanding the effects of FSI storage on the freshwater balance of both springsheds and surface watersheds.

*3.4. Hydrogeologic Settings*

In this study, 11 of the 55 basins (20%) evaluated for FSI-impacted water budgets had Δ*t* > 10 years and *S* > 0.4, indicating that the storage mechanism in these 11 basins was greater than can be explained by specific yield. Based on the characteristics of these basins shown in Table 2, the only identifiable trend with Δ*t* was a positive relationship with FSI depth (Figure 7). Linear regression between these two variables led to a slope of 0.016 yr/m with $R^2 = 0.30$, suggesting that a deeper FSI position leads to longer Δ*t*. Confining layers cover >50% of the drainage areas in all 11 basins. These low permeability units typically have a 1–4 order of magnitude difference between confining unit and aquifer permeability (based on Tables 9 and 13 from Williams and Kuniansky [21]). Basins in the SSI category typically showed stronger confining conditions, with >85% areal coverage of surficial confining units in four out of five basins, compared to two out of six SI basins. Despite these hydrogeologic differences, estimated Δ*t* in both categories were of the same order of

magnitude, which suggests that the FSI could be responsive to multidecadal changes in $R$ under both confined and unconfined conditions.

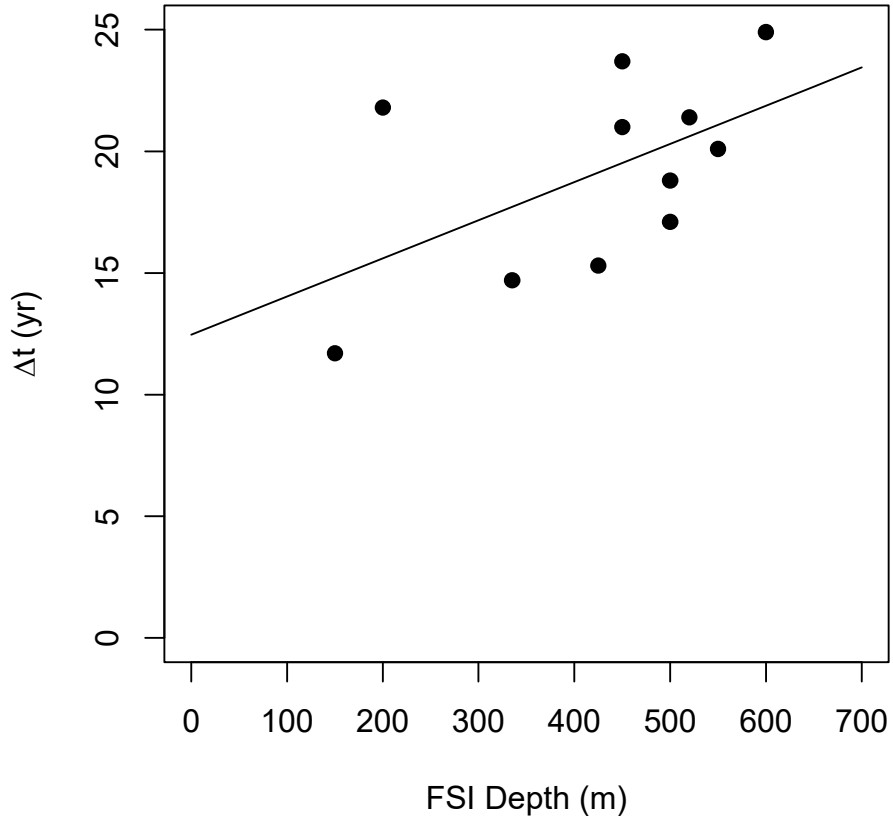

**Figure 7.** Linear regression between $\Delta t$ and FSI depth shows a slope of 0.016 yr/m with $R^2 = 0.30$.

Of the thirteen SSI basins, only the five listed in Table 2 showed $\Delta t > 10$ years. These five basins overlie aquifers with similar hydrogeology (Figure 6) of greater hydraulic resistance between the near-surface and the FSI than in an unconfined setting, in the form of either an upper confining unit or a non-leaky middle confining unit (MCU) (Table 2). The high $\Delta t$ and $S$ values in these five basins suggest the FSI responds to variable recharge to increase storage despite the presence of low permeability layers. Still important to note is that the longest period of record for stream discharge data in these basins is 80 years, the same length as one typical AMO cycle [15]. Even longer records would help improve confidence in estimated $\Delta t$ and $S$, but such records are currently not available and further investigation will depend on future studies.

Unlike in the five SSI basins that showed a $\Delta t$, the six SI basins showing a $\Delta t$ overlie areas of the FAS with varying hydrogeologic structure including confined (Basin 49), semi-confined (Basins 39 and 59), and unconfined conditions (Basins 40, 48, 63). Basins 39, 49, and 59 could each be expected to have shorter hydraulic response times in comparison to basins 40, 48, and 63 due to their small drainage areas (Table 2), but smaller values of transmissivity around these three basins [21] increases hydraulic response time. Thus, the estimation of $\Delta t$ may require more precise estimation of hydrogeologic parameters such as those shown in Equation (1). The variety of hydrogeologic conditions shown in Figure 6 and Table 2 for these six basins further supports the conclusion that the FSI could impact the water balance of both springsheds and watersheds in cases with complex hydrogeologic structures.

## 4. Conclusions

This study estimated both freshwater storage coefficients and $\Delta t$ between recharge $R$ and base flow $Q_b$ in 68 drainage basins impacted by AMO to assess the potential for

FSI impacts on freshwater budgets in the coastal southeastern United States. The FSI may impact freshwater budgets by providing a storage mechanism related to the Ghyben-Herzberg relationship that leads to delays in the response of groundwater discharge to aquifer recharge. This storage mechanism is "hidden" in the sense that it involves the movement of the FSI, which is not apparent from typical observations of changes in water table elevation. Only considering perennial stream or spring discharge stations in Florida and coastal Georgia with greater than 50 years of mostly continuous data, the 13 basins lacking evidence of a saltwater presence had $S < 0.4$, supporting the conclusion that their water budgets were not influenced by FSI, and 11 of the remaining 55 stations (20%) had both a $\Delta t$ between multidecadal $R$ and $Q_b$ trends in the range of 11–25 years and $S$ greater than can be explained by specific yield indicating that discharge lags on the multidecadal scale are attributed to the FSI movement. These 11 basins all overlie areas of the FAS with low permeability layers between the FSI and the water table and also showed a positive linear relationship between $\Delta t$ and FSI depth. Four of these basins did not show $\Delta t$ in $Q_s$ indicating that long-term surface runoff could conceal multidecadal $\Delta t$ in $Q_b$, and consequently obscure potential long-term trends in subsurface water quality. Therefore, the separation of surface runoff and $Q_b$ is necessary when interpreting multidecadal water budgets and solute mass balance in drainage basins connected to coastal aquifers interacting with saltwater.

The results presented in this study suggest that large time lags between recharge and groundwater discharge due to dynamic FSI behavior may be impacting both freshwater availability and resulting advective solute delivery to streams in the southeastern US. These concepts can improve the management of coastal freshwater resources by using the integration illustrated in Figure 1C to plan periods of reinforced aquifer recharge during low storage and increased groundwater withdrawal during high storage. Numerical modelling of the FSI response to multidecadal signals in $R$ is a logical next step for examining the hydrogeologic conditions that can support $\Delta t$ between aquifer recharge and discharge induced by the slow response of the FSI. Future studies investigating these potential impacts either in the southeastern US or elsewhere should use discharge time series that ideally at least match the length of one cycle of long-term periodicity in rainfall between positive and negative phases.

**Supplementary Materials:** The following supporting information can be downloaded at: https://www.mdpi.com/article/10.3390/w15010142/s1, Table S1: Statistics showing the quantity of missing values and the percentage of groundwater contribution to streamflow in the two basins; Table S2: Estimated $\Delta t$ statistics for the 11 sets of $R$ and $Q_b$ time series between all 3 basins; Figure S1: Time series of $R$ (black) and $Q_b$ (red) for each basin both prior to moving average window (top row) and after moving average window (bottom row). A moving average window of 20 years was used on the bottom row, and the lagged $Q_b$ time series is shown in dark red. The period-of-record mean was subtracted from each time series before plotting; Table S3: Estimated values of $S$ for each basin along with the resulting NSE between the best fit $\Delta h$ time series and the $\Delta V$ time series; Figure S2: Time series of $\Delta V$ (red) and $\Delta h$ (black) with best fit $\Delta h$ plotted as a dotted black line for the two basins. The solid black line multiplied by $S$ gives the dotted black line. NSE was estimated between the best fit $\Delta h$ time series and the $\Delta V$ time series. Reference [55] is listed in Supplementary Materials file.

**Author Contributions:** Conceptualization, H.K. and J.W.J.; methodology, B.E.; formal analysis, B.E.; investigation, M.D.A.; writing—original draft preparation, B.E.; writing—review and editing, J.W.J.; supervision, M.D.A.; project administration, J.W.J. and M.D.A. All authors have read and agreed to the published version of the manuscript.

**Funding:** Partial funding for Brady Evans through a University of Florida Graduate Research Fellowship.

**Data Availability Statement:** Data sharing not applicable. No new data were created or analyzed in this study. Data sharing is not applicable to this article.

**Conflicts of Interest:** The authors declare no conflict of interest.

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
