# Peer review of "Rainfall-Runoff Time Lags from Saltwater Interface Interactions in Atlantic Coastal Plain Basins"

_water, doi:10.3390/w15010142_

Round 1

Reviewer 1 Report

The manuscript  describes the results  obtained  by analyzing  hydrogeological  data  from 68  drainage basins  to assess the influence of FSI dynamics  on freshwater budgets in the coastal south-eastern United States. Particularly,  Authors found that a large time lag between recharge and groundwater discharge  may have high impacts  on both freshwater availability  and water quality of the surface water system.

 The paper is well written and   nicely  organized.  I only have some minor comments   reported in the attached pdf  file (yellow highlights).

Author Response

Reviewer 1

We thank this reviewer for their positive comments and suggested edits.

As part of revisions to the introduction suggested by another reviewer, the first appearance of the acronym “LFA” was moved to line 133 where it was specified. Additionally, the symbol “n” was defined in line 169.

The water budget equality in the methods section was explained and defined as new equations (Eq. 2 and Eq. 3).

We have confirmed the value of 36% of basins with Δt < 1 yr, and note that Figure 3 shows cdf = 0.36 (i.e., 36%) at Δt = 1 yr.

Reviewer 2 Report

The submitted paper titled “Rainfall-runoff time lags from saltwater interface interactions in Atlantic coastal plain basins” provides research about the freshwater-saltwater interface impacts at the watershed scale in regions of the United States. The manuscript is well-written however needs some changes. A lot of details about the hydrogeological regime of the study area are missing. I would like to mention some comments for its improvement and future publication in Water.

My recommendation is major revisions.

General comments

-Always mentioned in the third person.

-Abstract needs to be improved. Simplify and highlight the most important results.

-Introduction is very poor. Provide more information about climate variability, human needs, the importance of managed aquifer recharge, previews and new techniques about rainfall-runoff time lags etc.

-Figure 2. Add a coordinate system.

-Provide geological and/or hydrogeological cross-sections.

Suggested literature

Simulating future groundwater recharge in coastal and inland catchments. Water Resource Management.

Author Response

Reviewer 2

We thank this reviewer for their comprehensive suggestions for improving the manuscript.

All instances of first person language have been replaced with third person.

The abstract has been edited to be clearer and more concise. We are presenting the same results as in the previous version of the abstract because each result is important in characterizing the importance of the study or provides a direction for future research.

Coordinate systems have been added to both map figures (Figure 2 and Figure 6). Upon replacing these figures, Figures 3, 4, and 7 were also replaced with pdf images to preserve the intended symbology on each axis.

To address the lack of hydrogeological information in the manuscript, we have added basic hydrogeologic information for the study area including an in-text description of a typical hydrogeological cross section in the study area (lines 263-276). We then referred the reader to two publications for cross-sections in specific regions of the study area.

The introduction was the most heavily edited section due to the suggestions made by this reviewer. We have added “human needs” based motivation for the work in lines 43-48. Lines 49-60 were edited to contain more language on climate variability and the implications for recharge, discharge, and FSI movement. The next four paragraphs were largely unchanged because the concepts discussed in this manuscript must be meticulously explained as they are mostly new to the literature and critical in understanding the results of this manuscript. Additional information in the introduction beyond that in the current version is less important with regard to understanding the results. However, we agree that “managed aquifer recharge” is an important application of this work, and we added this statement in the conclusions (lines 485-487).

Reviewer 3 Report

This is a well written paper which clearly outlines the methodology used and the results. The analysis is sound and the conclusions approprite. 

I could not identify any minor errors.

Author Response

Reviewer 3

We thank this reviewer for their positive comments on the manuscript.

Round 2

Reviewer 2 Report

I have read the revised version of the manuscript which has covered my initial comments.